# PATCHREFINER V2: FAST AND LIGHTWEIGHT REAL-DOMAIN HIGH-RESOLUTION METRIC DEPTH ESTIMATION

**Zhenyu Li, Wenqing Cui, Shariq Farooq Bhat, Peter Wonka**
KAUST
`{zhenyu.li.1,wenqing.cui,shariq.bhat,peter.wonka}@kaust.edu.sa`

## ABSTRACT

While current high-resolution depth estimation methods achieve strong results, they often suffer from computational inefficiencies due to reliance on heavyweight models and multiple inference steps, increasing inference time. To address this, we introduce PatchRefiner V2 (PRV2), which replaces heavy refiner models with lightweight encoders. This reduces model size and inference time but introduces noisy features. To overcome this, we propose a Coarse-to-Fine (C2F) module with a Guided Denoising Unit for refining and denoising the refiner features and a Noisy Pretraining strategy to pretrain the refiner branch to fully exploit the potential of the lightweight refiner branch. Additionally, we propose to adopt the Scale-and-Shift Invariant Gradient Matching (SSIGM) loss within local windows to enhance synthetic-to-real domain transfer. PRV2 outperforms state-of-the-art depth estimation methods on UnrealStereo4K in both accuracy and speed, using fewer parameters and faster inference. It also shows improved depth boundary delineation on real-world datasets like CityScapes, demonstrating its effectiveness.

## 1 INTRODUCTION

Accurate high-resolution depth estimation from a single image is critical for advancements in fields such as autonomous driving, augmented reality, and 3D reconstruction Eigen et al. (2014); Zhang et al. (2023); Bhat et al. (2023); Li et al. (2024c). Current state-of-the-art depth estimation models typically operate at relatively low resolutions (e.g., 0.3 megapixels). High memory requirements, especially at 4K resolution, pose a significant challenge for training depth estimation models that can natively support high-resolution inputs. Recent 4K depth estimation approaches like PatchRefiner Li et al. (2024b) (PRV1) use a tile-based strategy where the high-resolution image is divided into patches. The patch-level depth predictions (fine, local outputs) are then fused with the depth prediction of a downsampled version of the input image (coarse, global output) to obtain a single, consistent, high-resolution output.

Despite its success, the PatchRefiner framework faces critical computational efficiency and scalability challenges for real-world applications. It employs the same architecture (a pre-trained base depth model) to extract both global and patch-level features. This amounts to at least 16 forward passes of the base model for a single high-resolution input. As the base model used Bhat et al. (2023); Yang et al. (2024); Yang et al. is often large, this results in two major issues: 1) **High inference time** of more than a second per image, and more importantly 2) **High memory requirement**, making the end-to-end training infeasible. Therefore, the PRV1 framework has to adopt stage-wise training, where global and local branches are trained sequentially, leading to a long training time and suboptimal results.

To alleviate these issues, we propose to substitute the large foundational models, such as ZoeDepth Bhat et al. (2023) or DepthAnything Yang et al. (2024); Yang et al., used in the refiner branch Li et al. (2024b) with lightweight encoders like MobileNet Howard et al. (2019); Qin et al. (2024) and EfficientNet Tan & Le (2019). This change significantly reduces the number of parameters and memory usage, decreases inference time, and enables end-to-end training without bells and whistles. However, this modification introduces a trade-off: the model capacity is reduced, and

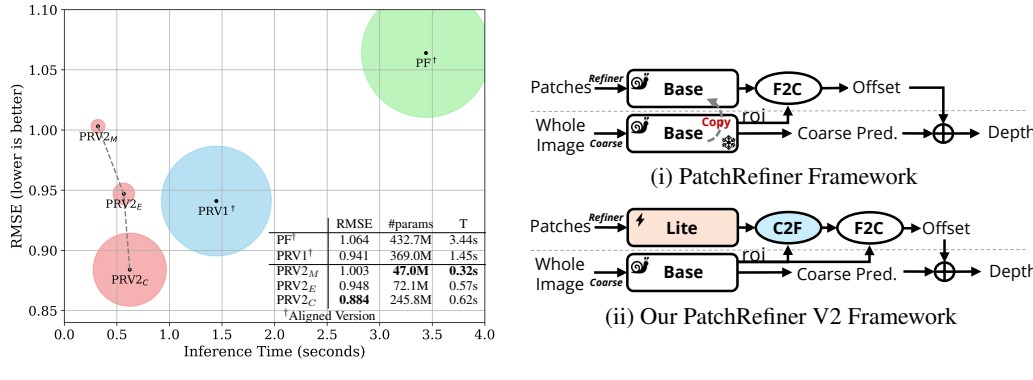

(a) Comparison on UnrealStereo4K.

(b) A Comparison of PRV1 and PRV2.

Figure 1: PatchRefiner V2 (PRV2) significantly outperforms previous high-resolution frameworks. PF and PRV1 are short for PatchFusion Li et al. (2024a) and PatchRefiner Li et al. (2024b), respectively. We adopt a **lightweight encoder** for the refiner branch, which alleviates the inference speed bottleneck, reduces the number of parameters for high-resolution estimation, and facilitates end-to-end training. A novel **coarse-to-fine (C2F)** module is proposed to denoise features from the lite model and further boost performance.

the refiner branch now lacks the depth-aligned feature representation otherwise provided by the previously used pre-trained depth estimation base models. While end-to-end training alleviates some of this limitation, the lack of depth-aligned feature representation remains a concern. Indeed, we observe that the features generated by these lightweight encoders tend to be 'noisy' (see Fig. 2) even after ImageNet initialization Deng et al. (2009) and end-to-end training. This causes the original Fine-to-Coarse module (F2C) used in PRV1 to struggle to inject rich, high-resolution information for the final depth prediction.

We propose two components to improve the feature representation in the refiner branch: 1) The **Coarse-to-fine module (C2F)**, which incorporates novel Guided Denoising Units (GDUs) in a bottom-to-top manner Lin et al. (2017); Xian et al. (2018); Ranftl et al. (2021). GDUs utilize coarse depth features as guidance to denoise and enhance the high-resolution refiner features. Together with the original Fine-to-Coarse module (F2C), this establishes a bidirectional fusion process: C2F initially denoises and refines high-resolution features using coarse features, followed by F2C's enhancement of the predicted coarse depth map via residual prediction. 2) **Noisy Pre-training.**[1] Given that the C2F and F2C modules require initialization from scratch, we propose a simple pre-training strategy for the entire refiner branch — including the encoder, C2F, and F2C modules — to enhance feature representation and accelerate learning. During Noisy Pre-training, we replace the input coarse depth features for GDUs with random noise, essentially forcing the refiner branch to learn to extract depth-relevant features from the high-resolution input.

Finally, the PRV1 framework Li et al. (2024b) employs the Detail and Scale Disentangling (DSD) training strategy to adapt the high-resolution depth estimation framework to real-domain datasets, which enables learning 'detail' from synthetic data and 'scale' from the real domain. To isolate the scale from the synthetic data, the DSD strategy uses a ranking loss and Scale-and-Shift Invariant (SSI) loss Ranftl et al. (2022). To further improve the transfer of high-frequency knowledge, we introduce a gradient-level loss Li & Snavely (2018); Ranftl et al. (2022) applied after scale-and-shift alignment within *local* windows Bhat et al. (2022); Wang et al. (2025), which we term the local Scale-and-Shift Invariant Gradient Matching (*local SSIGM*) loss. While the formulation of the gradient loss follows Li & Snavely (2018); Ranftl et al. (2022), our method differs in two key aspects: (1) supervision is derived from pseudo labels generated by a teacher model trained on a synthetic dataset, and (2) the loss is computed within local windows rather than across the entire depth map. These modifications are designed to mitigate potential distortions in accurate scale estimation and to encourage the model to focus on fine-grained local structures.

---

[1]We use the term 'pre-training' loosely, as this process occurs prior to the final training phase.

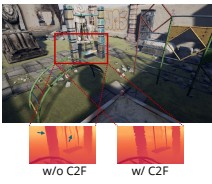 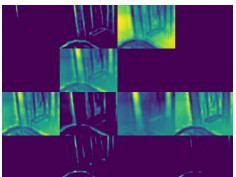 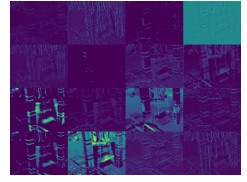 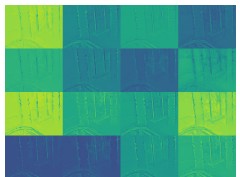

(a) Input, Prediction    (b) Coarse Feature    (c) Refiner Feature (w/o C2F) (d) Refiner Feature (w/ C2F)

Figure 2: **Visualization of F2C Input Feature Maps.** We showcase the first 16 channels of the F2C input features. (c) Without the C2F module (setting ③ in Tab. 2a), the refiner features are 'noisy' and hard to interpret. (d) The C2F module helps denoise the refiner features, leading to clear boundaries and better results.

Experiments demonstrate that our advanced framework, **PatchRefiner V2** (PRV2), performs effectively across various lightweight architectures. As summarized in Fig. 1a, PRV2 significantly outperforms other high-resolution metric depth estimation frameworks on the UnrealStereo4K Tosi et al. (2021) dataset in terms of both quantitative results and inference speed. Additionally, we evaluate the effectiveness of our local SSIGM loss on the real-world dataset CityScapes Cordts et al. (2016). Our method reveals significant improvements in depth boundary delineation (e.g., +25.1% boundary F1 *w.r.t* Li et al. (2024b)) while maintaining accurate scale estimation, showcasing its adaptability and effectiveness.

## 2 RELATED WORK

### 2.1 HIGH-RESOLUTION MONOCULAR DEPTH ESTIMATION

Monocular depth estimation (MDE) is a fundamental computer vision task and has recently seen impressive progress with advanced network design Eigen et al. (2014); Bhat et al. (2021); Li et al. (2023; 2024c); Bhat et al. (2023); Yang et al. (2021), supervision Lee & Kim (2020); Liu et al. (2023a); Xian et al. (2020); Ranftl et al. (2022); Godard et al. (2019), formulation Fu et al. (2018); Diaz & Marathe (2019); Bhat et al. (2021); Xian et al. (2020); Li et al. (2024c); Bhat et al. (2022), training strategy Petrovai & Nedevschi (2022); Godard et al. (2019); Fan et al. (2023); Ranftl et al. (2022), public datasets Silberman et al. (2012); Geiger et al. (2013); Dai et al. (2017); Cordts et al. (2016); Roberts et al. (2021), *etc*. Recently, most SOTA frameworks Bhat et al. (2023); Yang et al. (2024); Yang et al.; Ke et al. (2024) build on the top of heavy backbones Bao et al. (2022); Dosovitskiy et al. (2021); Oquab et al. (2025); Rombach et al. (2022), leading to the limitation of low-resolution input. For example, Depth Anything V2 Yang et al. uses ViT-L Dosovitskiy et al. (2021); Oquab et al. (2025) and can only infer 756×994 (about 0.75 megapixels) images on an NVIDIA V100 32G GPU. Another recent work, DepthPro Bochkovskii et al. (2025), presents a high-resolution framework whereas the input resolution is fixed at 1536. While another line of research utilizing the generative model for MDE achieves fine-grained results Ke et al. (2024); Pham et al. (2025); Xu et al. (2025), a similar dilemma exists. For instance, Marigold Ke et al. (2024) based on Stable Diffusion Rombach et al. (2022) runs with ∼0.33 megapixels as default.

This contrasts with the advancements in modern imaging devices, which increasingly capture images at higher resolutions, reflecting the growing demand for high-resolution depth estimation Li et al. (2024a). To relax the constraints, initial efforts utilize Guided Depth Super-Resolution (GDSR) Metzger et al. (2023); Hui et al. (2016); Zhong et al. (2023) and Implicit Functions Mildenhall et al. (2021); Chen et al. (2021). Recent works adopt a tile-based method to segment images into patches for estimation and then reassemble all predictions into a comprehensive depth map Miangoleh et al. (2021); Li et al. (2024a;b); Kwon & Kim (2025). Since all these methods adopt the dual-branch architecture and utilize the same SOTA depth model in both branches, the frameworks are heavy and slow at inference time. By contrast, we aim to achieve fast, high-resolution metric depth estimation using the tile-based method with fewer additional parameters. More specifically, Kwon & Kim (2025) proposes the grouped patch consistency training and bias-free masking to improve patch consistency and mitigate dataset-specific biases. Their approach focuses on consistency learning, which is orthogonal and complementary to our focus on lightweight architectural design for efficient high-resolution depth estimation.

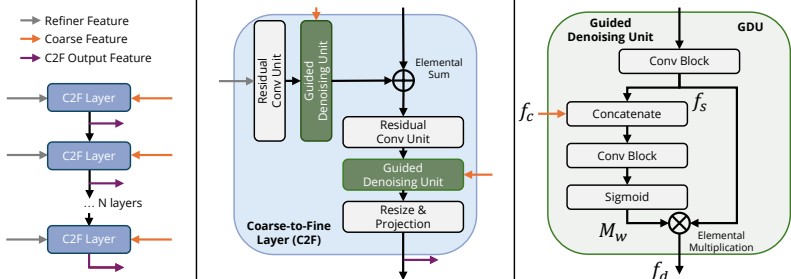

Figure 3: *Left*: Coarse-to-Fine (C2F) module overview. It processes refiner features in a bottom-to-top manner with $N$ successive C2F layers. Each layer is guided by coarse features with corresponding resolution and outputs denoised features for the Fine-to-Coarse (F2C) module. *Center*: C2F layers combine multi-level features with Residual Convolutional Units Lin et al. (2017); Ranftl et al. (2022) and denoises the features using Guided Denoising Units (GDU). *Right*: Guidance information from the coarse branch is introduced through a concatenation followed by a convolutional block and then converted to a weight map ranging from 0 to 1 through the sigmoid operator. We then adopt an elementwise multiplication to denoise the shortcut feature.

## 2.2 SYNTHETIC-TO-REAL TRANSFER FOR MDE

The challenge of obtaining high-quality, real-domain data for training high-resolution depth models has led recent efforts to utilize synthetic datasets, thereby encountering significant domain gaps during real-world inference Rajpal et al. (2023); Li et al. (2024a). To address this issue, PatchRefiner combines labeled data from both synthetic and real domains, enhancing depth estimation in real-world, high-resolution settings Li et al. (2024b). Inspired by the successes of semi-supervised learning Van Engelen & Hoos (2020); Yang et al. (2024); Kirillov et al. (2023), it employs a pseudo-labeling approach Pseudo-Label (2013); Saito et al. (2017); Chen et al. (2019); Pastore et al. (2021); Shin et al. (2022) along with the Detail and Scale Disentangling (DSD) loss. This strategy facilitates the transfer of fine-grained knowledge from synthetic to real domains. In our work, we extend this concept by incorporating supervision in the gradient space and within local windows, thereby significantly enhancing the effectiveness of knowledge transfer.

## 3 METHOD

### 3.1 REVISITING PATCHREFINER

We first revisit the PatchRefiner framework Li et al. (2024b) (named as PRV1). The PRV1 framework adopts a tile-based approach to address the high memory and computational demands of high-resolution depth estimation Li et al. (2024a); Miangoleh et al. (2021). It utilizes a two-step process: **(i)** Coarse Depth Estimation and **(ii)** Fine-Grained Depth Refinement, as shown in Fig. 1b.

**(i) Coarse Depth Estimation:** This step involves a coarse depth estimation network, $\mathcal{N}_c$, which processes downsampled inputs to generate a global depth map, $\mathbf{D}_c$. This map captures the overall scene structure and provides a baseline for further refinement. Notably, $\mathcal{N}_c$ can be any depth estimation model and is kept fixed after this stage.

**(ii) Fine-Grained Depth Refinement:** PRV1 introduces a unified refinement network, $\mathcal{N}_r$, in place of separate fine depth networks and fusion mechanisms Li et al. (2024a); Poucin et al. (2021). This network refines the coarse depth map by recovering details and enhancing depth precision at a patch level.

The refinement process begins with the cropped input image $I$, processed by a base depth model $\mathcal{N}_d$, which shares the same architecture as $\mathcal{N}_c$. Multi-scale features from both $\mathcal{N}_d$ and $\mathcal{N}_c$ are collected as $\mathcal{F}_d = \{f_d^i\}_{i=1}^L$ and $\tilde{\mathcal{F}}_c = \{\tilde{f}_c^i\}_{i=1}^L$. Following Li et al. (2024a), the `roi` He et al. (2017) operation extracts features from the cropped area as $\tilde{f}_c^i = \texttt{roi}(f_c^i)$.

These features are then aggregated by a lightweight decoder through concatenation and convolutional blocks, referred to as the Fine-to-Coarse (F2C) module in this paper, which injects fine-grained information into the coarse refinement process. The F2C module constructs the residual depth map $\mathbf{D}_r$ at the input resolution, and the final patch-wise depth map is computed as $\mathbf{D} = \texttt{roi}(\mathbf{D}_c) + \mathbf{D}_r$.

As the second contribution, PRV1 introduces a teacher-student framework to transfer the fine-grained knowledge learned from the synthetic data to the real domain. The Detail and Scale Disentangling (DSD) loss is designed to help the model balance detail enhancement with scale accuracy by integrating both the scale-consistent ground truth supervision and the detail-focused pseudo labels. Both ranking loss Xian et al. (2020) and the scale-and-shift invariant loss Ranftl et al. (2022) can be adopted for pseudo-label supervision.

**Limitations of PRV1.** Similar to other tile-based methods Poucin et al. (2021); Li et al. (2024a), the PRV1 framework encounters significant challenges with the computational efficiency and scalability for real-world applications due to the shared usage of the base depth model (e.g., ZoeDepth Bhat et al. (2023), Depth Anything Yang et al. (2024); Yang et al.) across both the coarse and refiner branches. For a given input image, while the coarse branch processes the downsampled image once to gather global information, the refiner branch requires multiple inferences (at least 16 in PRV1's default mode) for the patches. Since both branches share the same architecture, the refiner branch becomes the primary efficiency bottleneck. Our goal is to alleviate this bottleneck as much as possible.

Moreover, a heavy framework makes end-to-end training infeasible due to GPU memory limitations. The PRV1 framework has to adopt two stages for training the framework, where global and local branches are trained sequentially. This results in a long training time and suboptimal performance. While the authors claim that multiple-stage training could potentially lead to stage-wise local optima Li et al. (2024b), our goal is to pursue end-to-end training.

## 3.2 PATCHREFINER V2 FRAMEWORK

### 3.2.1 LITE FRAMEWORK FOR FASTER INFERENCE AND END-TO-END TRAINING

We propose a simple solution to address PRV1's limitations: a lightweight architecture for the refiner branch. Given that the coarse branch already provides a reliable base depth estimation $\mathbf{D}_c$, using the same heavy model for the refiner might be unnecessary. This substitution significantly increases inference speed, reduces the model size, and enables end-to-end training. However, it also results in a noticeable decline in refinement quality compared to previous methods Li et al. (2024a;b). We attribute this decline to the lack of depth-aligned feature representation in the refiner branch, as shown in Fig. 2.

To compensate for the loss in model capacity and depth-pretraining by the proposed substitution, we introduce a better architecture design, a Coarse-to-Fine (C2F) Module, and a fast and simple pre-training strategy, Noisy Pretraining (NP).

### 3.2.2 COARSE-TO-FINE MODULE

Since the refiner branch no longer includes a pretrained depth model, we propose utilizing information from the global coarse branch to guide the selection of relevant details from the fine, patch-level features.

The proposed Coarse-to-Fine (C2F) module shown in Fig. 3 processes the multi-scale features extracted from the lightweight encoder through $N$ successive C2F layers in a bottom-to-up manner Ronneberger et al. (2015); Lin et al. (2017), mirroring the design of the Fine-to-Coarse (F2C) module Li et al. (2024b). Each C2F layer is designed to progressively enhance and denoise the refiner features with the help of coarse feature representations.

Each layer in the C2F module consists of two components: our proposed Guided Denoising Unit (GDU) and the Residual Convolutional Unit Lin et al. (2017); Ranftl et al. (2022). The GDU introduces coarse feature maps $f_c$ at each stage to refine and denoise the refiner features. Specifically, the coarse features serve as guidance, which are incorporated via the concatenation operation ($\texttt{Cat}$)

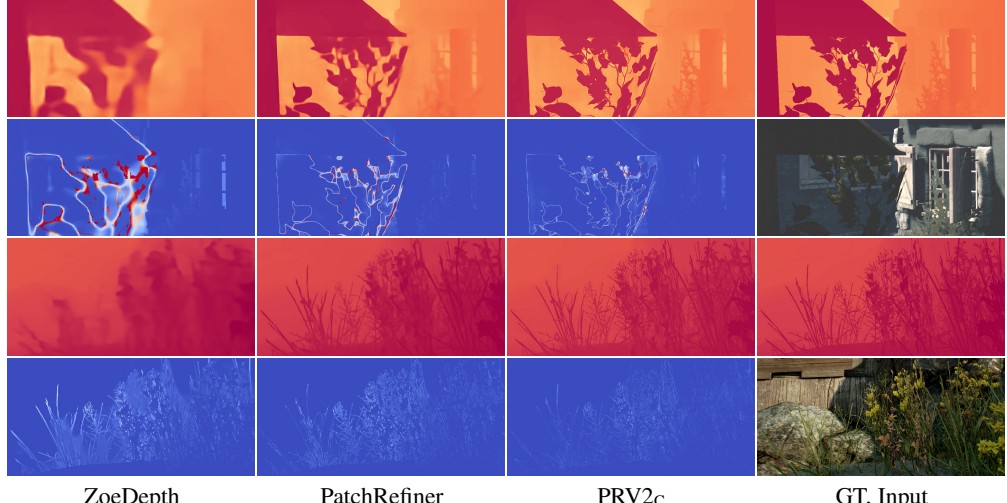

|           |              |          |           |
|-----------|--------------|----------|-----------|
| ZoeDepth  | PatchRefiner | PRV2$_C$ | GT, Input |

Figure 4: **Qualitative Comparison on UnrealStereo4K.** We show the depth prediction and corresponding error map, respectively. The qualitative comparisons showcased here indicate our PRV2$_C$ outperforms counterparts Bhat et al. (2023); Li et al. (2024b) with sharper edges and lower error around boundaries while achieving faster inference. We show ***individual patches*** in all images to emphasize details near depth boundaries.

followed by the convolutional block (CB). The output of these blocks is passed through a sigmoid activation function ($\sigma$) to obtain a weight map $M_w$, which ranges from 0 to 1. This weight map is then applied to the shortcut features $f_s$ through elemental multiplication $\otimes$, effectively denoising the shortcut features. This process can be formulated as

$$M_w = \sigma(\texttt{CB}(\texttt{Cat}(f_c, f_s))), \tag{1}$$

$$f_d = M_w \otimes f_s, \tag{2}$$

where the $f_d$ indicates the denoised feature. Associated with the Residual Convolutional Unit, it allows the model to filter out irrelevant noise and enhance the quality of the refined features iteratively across the network layers. After that, we utilize the F2C module to inject the denoised fine-grained information for coarse features, leading to a more effective and better refinement process.

### 3.2.3 NOISY PRETRAINING

In PRV1, the framework's efficacy largely depends on the comprehensive pretraining of the base models in both the coarse and refiner branches Li et al. (2024b). During the subsequent high-resolution training stage, only the Fine-to-Coarse (F2C) module is trained from scratch, representing a minor portion of the overall refiner branch (24.0M *vs.* 369.0M parameters). In other words, a significant portion ($\sim$94%) of the refiner branch is pretrained for depth estimation.

By our substitution, this pretraining is also lost. While the lightweight encoder used can be pretrained on a large-scale dataset with complex strategies, it now constitutes only a small part of the refiner branch (1.3M *vs.* 47.0M parameters for PRV2$_M$) and lacks the depth-aligned feature representation. In other words, even if we pre-train the encoder, a significant portion ($\sim$98%) of the refiner branch must still be trained from scratch.

To address this issue, we propose a novel approach called Noisy Pretraining (NP). Prior to the high-resolution training, we pretrain the lightweight encoder along with the C2F and F2C modules. However, a critical aspect of our framework is that both the C2F and F2C modules rely on features from the base model in the coarse branch. These features, however, are challenging to omit during the pretraining process. We propose a straightforward yet effective solution: we randomly generate the coarse features using a normal distribution $N(0, 1)$ as inputs. This forces the refiner branch to learn depth-relevant features without guidance from the coarse branch.

Table 1: **Quantitative Comparison on UnrealStereo4K.** Best results are marked **bold**. PF, PRV1 and PRV2 are short for PatchFusion Li et al. (2024a), PatchRefiner Li et al. (2024b) and PatchRefiner V2, respectively. We report the $P = 16$ mode for these high-resolution depth estimation frameworks Li et al. (2024a). Gray lines present numbers from the original paper with vanilla pretraining settings. [†]: indicates the pretraining aligned version, where we remove the **non-public** Midas pretraining stage Ranftl et al. (2022) adopted for the *fine or refiner branch* in PRV1 and PF to make fair comparisons with our PRV2. The coarse branch is **NOT** modified. #param. and T denote the number of additional parameters adopted for high resolution estimation and the inference time of the *fine or refiner branch* for one input image. Best results are in **bold**. SharpDepth Pham et al. (2025) is not involved in this benchmark as different training and evaluation protocols.

| Method | $\delta_1(\%)\uparrow$ | REL$\downarrow$ | RMSE$\downarrow$ | SiLog$\downarrow$ | SEE$\downarrow$ | #param$\downarrow$ | T$\downarrow$ | Reference |
|---|---|---|---|---|---|---|---|---|
| iDisc Piccinelli et al. (2023) | 96.940 | 0.053 | 1.404 | 8.502 | 1.070 | | | ICCV 2023 |
| SMD-Net Tosi et al. (2021) | 97.774 | 0.044 | 1.282 | 7.389 | 0.883 | - | - | CVPR 2021 |
| Graph-GDSR De Lutio et al. (2022) | 97.932 | 0.044 | 1.264 | 7.469 | 0.872 | | | CVPR 2022 |
| Boosting Miangoleh et al. (2021) | 98.104 | 0.044 | 1.123 | 6.662 | 0.939 | | | CVPR 2021 |
| ZoeDepth Bhat et al. (2023) | 97.717 | 0.046 | 1.289 | 7.448 | 0.914 | - | - | - |
| ZoeDepth+PF Li et al. (2024a) | 98.419 | 0.040 | 1.088 | 6.212 | 0.838 | 432.7M | 3.44s | CVPR 2024 |
| ZoeDepth+PF[†] Li et al. (2024a) | 98.369 | 0.039 | 1.064 | 6.342 | 0.855 | | | |
| ZoeDepth+PRV1 Li et al. (2024b) | 98.821 | 0.033 | 0.892 | 5.417 | 0.750 | 369.0M | 1.45s | ECCV 2024 |
| ZoeDepth+PRV1[†] Li et al. (2024b) | 98.680 | 0.034 | 0.941 | 5.614 | 0.771 | | | |
| ZoeDepth+**PRV2**$_M$ | 98.610 | 0.034 | 1.003 | 5.760 | 0.832 | **47.0M** | **0.32s** | |
| ZoeDepth+**PRV2**$_E$ | 98.728 | 0.034 | 0.948 | 5.579 | 0.816 | 72.1M | 0.57s | **Ours** |
| ZoeDepth+**PRV2**$_C$ | **98.863** | **0.032** | **0.884** | **5.281** | 0.787 | 245.8M | 0.62s | |

Unlike other strategies Liu et al. (2023b); Ozguroglu et al. (2024); Brooks et al. (2023), which often require careful selection and modification of convolutional layers and their corresponding parameters, our NP method avoids altering the framework's architecture. As a result, the pretraining and subsequent training stages proceed seamlessly, preserving the integrity of the overall model structure while ensuring that all components of the refiner branch are well-prepared for high-resolution training.

## 3.3 LOCAL SCALE-AND-SHIFT INVARIANT GRADIENT MATCHING

In the synthetic-to-real transfer stage, PRV1 employs the scale-and-shift invariant (SSI) loss $\mathcal{L}_{\text{SSI}}$ as the pseudo-label supervision within the Detail and Scale Disentangling (DSD) loss. To better transfer high-frequency information, we extend the supervision to the gradient domain and compute it *locally*.

Let $d$ be the predicted depth and $\hat{d}$ the pseudo label generated by a teacher trained on synthetic data. For each training patch, we randomly sample $N$ square windows $\{\Omega_k\}_{k=1}^N$ of side length $\ell$. For each window we estimate a *local* scale–shift pair $(s_k, t_k)$ by least-squares alignment Ranftl et al. (2022):

$$(s_k, t_k) = \arg\min_{s,t} \sum_{p \in \Omega_k} \left( sd(p) + t - \hat{d}(p) \right)^2. \tag{3}$$

Our local Scale-and-Shift Invariant Gradient Matching (local SSIGM) can be formulated as

$$\mathcal{L}_{\text{local-SSIGM}} = \frac{1}{N} \sum_{k=1}^N \frac{1}{|\Omega_k|} \sum_{p \in \Omega_k} \left( |\nabla_x r_k(p)| + |\nabla_y r_k(p)| \right), \tag{4}$$

where $\nabla_x$ and $\nabla_y$ are finite-difference gradients as in Li & Snavely (2018). Here, $r_k(p)$ is the residual of aligned prediction and ground-truth depth, calculated by $r_k(p) = s_k d(p) + t_k - \hat{d}(p), p \in \Omega_k$. The loss is combined with a weight $\lambda$ to control the strength of pseudo-label supervision, as in PRV1 Li et al. (2024b).

By aligning and matching gradients *within* windows rather than over the entire map, local SSIGM reduces the influence of global scale biases and forces the model to focus on fine-grained structures (*e.g.*, boundaries and thin objects) during high-frequency knowledge transfer. Note that setting $N=1$ and $\Omega_1$ to the full image recovers the original (global) SSIGM.

# 4 EXPERIMENTS

## 4.1 DATASETS AND METRICS

We evaluate the effectiveness of our proposed framework on the UnrealStereo4K dataset Tosi et al. (2021) (Synthetic), which offers synthetic stereo images at a 4K resolution ($2160\times3840$), each paired with accurate, boundary-complete pixel-wise ground truth. Adhering to the dataset splits in Tosi et al. (2021), we employ a default patch size of $540\times960$ for compatibility with Li et al. (2024b). In terms of the synthetic-to-real transfer part, we use the Cityscapes Cordts et al. (2016) dataset following PRV1 Li et al. (2024b). Following Li et al. (2024a), we adopt standard depth evaluation metrics from Eigen et al. (2014) and the Soft Edge Error (SEE) from Tosi et al. (2021) for *scale* evaluation. As for the real-domain datasets (Cityscapes), we adopt the standard protocol introduced in Li et al. (2024b) and utilize the F1 score to evaluate the boundary quality.

## 4.2 IMPLEMENTATION DETAILS

**PRV2 on Synthetic Dataset:** For training on the synthetic dataset, we employ the scale-invariant log loss $\mathcal{L}_{silog}$, as introduced in Eigen et al. (2014). We initialize the coarse network $\mathcal{N}_c$ with pretrained weights from the NYU-v2 dataset Silberman et al. (2012), adhering to the approach in Li et al. (2024a;b) for a fair comparison. As for the refiner branch, we employ the MobileNetV4-Small Qin et al. (2024), EfficientNet-B5 Tan & Le (2019), and Convnext-Large for $PRV2_M$, $PRV2_E$, and $PRV2_C$, respectively. We perform the noisy pretraining for the refiner branch for 96 epochs. The $\mathcal{N}_c$ is independently trained for 24 epochs and fine-tuned with the refiner branch in a fully end-to-end manner for another 48 epochs on the synthetic dataset. During inference, we use the Consistency-Aware Inference, as described in Li et al. (2024a).

**Learning on Real-Domain Dataset:** Following Li et al. (2024b), we first train the full $PRV2_E$ framework on the target real-domain dataset with the same setting as the synthetic dataset. During this stage, we perform an ablation on the NP strategy by toggling it on and off and reporting its impact on the final results. After that, we fine-tune the model with the Detail and Scale Disentangling loss for three epochs to refine depth estimations. The weight of the DSD loss is set as 0.8.

## 4.3 EXPERIMENTAL RESULTS AND DISCUSSION

**Main Results:** As shown in Tab. 1, our most lightweight model, $PRV2_M$, not only improves RMSE by 22.2% compared to the base depth model but is also 9.2x smaller and 10.7x faster than PatchFusion (PF) in terms of parameter count and inference speed, respectively. Our middleweight model, $PRV2_E$, achieves comparable RMSE to the previous SoTA PRV1 while being 2.5x faster and 5.1x smaller, offering an excellent balance between performance and efficiency. With ConvNext as the backbone, $PRV2_C$ sets a new SoTA with an RMSE of 0.884, while being 2.3x faster than PR. Qualitative results in Fig. 4 demonstrate $PRV2_C$'s superior boundary delineation.

We ablate and discuss the contributions of individual components proposed for PRV2. We employ the MobileNet in the refiner branch for experiments on the synthetic dataset and EfficientNet on the real-domain dataset. By default, we adopt $P = 16$ patches for clarity and ease of comparison.

**Framework Design:** As shown in Tab. 2a, we start with a baseline framework (①) in which we only substitute the base depth model in the refiner branch in PR with the lightweight encoder. While the inference time and model size are drastically reduced, quality is also degraded by a large margin. Simply adding more parameters to scale up the F2C cannot improve the performance, as shown in ②. Adopting an end-to-end training strategy can help improve the model performance (③). With the help of our proposed C2F that denoises the refiner features effectively, as shown in Fig. 2, the model's RMSE is reduced by 12.2% while only introducing a satisfactory overhead (③, ④). We also adopt different variants for C2F to evaluate the effectiveness. Firstly, we remove the GDU so that the C2F degrades to a simple bottom-to-top aggregation module (⑥). While it can still improve the model performance, there is a large margin compared with the complete C2F module. Then, we replace the GDU with the fusion module used in F2C (⑦). This results in a significant drop in performance. We argue this is due to the coarse features dominating the fusion process. The high-frequency information cannot be preserved correctly, leading to a performance on par with ①.

Table 2: **Ablation Study of Design Choice.** F2C and C2F denote the fine-to-coarse and coarse-to-fine module in the bi-directional fusion module, respectively. E2E, NP, GM, and *wins.* are short for end-to-end training, noisy pretraining, gradient matching, and local windows, respectively. Time: average inference time of the refiner branch for one image. ranking and SSI are the major contributions from PRV1 Li et al. (2024b).

(a) Ablation Study on UnrealStereo4K.

| | F2C | E2E | C2F | NP | Method | RMSE | #param. | T(s) |
|---|---|---|---|---|---|---|---|---|
| | | | | | Coarse Baseline | 1.289 | - | - |
| ① | ✓ | | | | | 1.201 | 27.5M | 0.08s |
| ② | ✓ | | | | | 1.214 | 70.2M | 0.38s |
| ③ | ✓ | ✓ | | | | 1.184 | 27.5M | 0.08s |
| ④ | ✓ | ✓ | ✓ | | | 1.041 | 47.0M | 0.32s |
| ★ | ✓ | ✓ | ✓ | ✓ | | 1.003 | 47.0M | 0.32s |
| ⑥ | | | | | w/o GDU | 1.137 | 34.5M | 0.19s |
| ⑦ | | | | | replace GDU with PatchRefiner fusion | 1.202 | 47.0M | 0.32s |
| ⑧ | | | | | NP, only load encoder | 1.029 | 47.0M | 0.32s |
| ⑨ | | | | | w/o ImageNet pretraining | 1.059 | 47.0M | 0.32s |

(b) Ablation Study on Cityscapes.

| | | | F2C | E2E | C2F | | Method | RMSE | F1 |
|---|---|---|---|---|---|---|---|---|---|
| | | | | | | | coarse baseline | 9.097 | 19.15 |
| | | | ✓ | | | | | 8.890 | 22.27 |
| | | | ✓ | ✓ | | | | 8.849 | 22.87 |
| | | | ✓ | ✓ | ✓ | | | 8.513 | 27.98 |
| | NP | ranking | SSI | DSD Loss GM | *wins.* | | | |
| ① | | | | | | | | 8.513 | 27.98 |
| ② | | ✓ | | | | | | 8.533 | 28.27 |
| ③ | | | ✓ | | | | | 8.533 | 29.22 |
| ④ | | | ✓ | ✓ | | | | 9.022 | 33.47 |
| ⑤ | ✓ | | ✓ | ✓ | | | | 8.534 | 35.32 |
| ★ | ✓ | | ✓ | ✓ | ✓ | | | 8.527 | 36.54 |

Table 3: **Ablation Study of Local SSIGM on Cityscpaes.** When varying window size, the number of windows is fixed to the best setting, and vice versa.

| *wins.* size | 5 | 11 | 23 | 47 | 95 | 191 |
|---|---|---|---|---|---|---|
| Variants RMSE / F1 | 8.528 / 34.28 | 8.523 / 36.04 | 8.527 / **36.54** | 8.525 / 36.06 | 8.532 / 35.79 | 8.538 / 35.65 |
| # of *wins.* | 0 | 20 | 50 | 100 | 200 | 300 |
| Variants RMSE / F1 | 8.534 / 35.32 | 8.523 / 36.43 | 8.532 / 36.52 | 8.527 / **36.54** | 8.529 / 36.51 | 8.529 / 36.49 |

**Noisy Pretraining:** When equipped with NP (★), our model achieves the best performance with 22.2% improvement over the coarse baseline in terms of RMSE. To prove our claim in the method section, we conduct the experiment by only loading the encoder part parameters after the NP process (⑧). The discrepancy in performance indicates that the pretraining of C2F and F2C modules is also crucial for the model performance, which is often ignored in the current depth estimation community. Then, we discard the ImageNet Deng et al. (2009) pretrained parameters for the lightweight encoder and train the entire refiner branch from scratch (⑨). The result validates our assumption that pretraining is crucial for the refiner. Moreover, as shown in Tab. 2b, the NP also plays a crucial role for training a real-domain high-resolution model (④, ⑤).

**Local SSIGM for Real-Domain Dataset:** Tab. 2b illustrates the performance gains achieved with our proposed local SSIGM loss. While maintaining a comparable scale RMSE to the ranking and SSI losses used in Li et al. (2024b), local SSIGM significantly improves boundary F1 scores, with gains of 25.1%. The detailed ablation study demonstrates the effectiveness of both applying the gradient matching after scale-and-shift alignment and the local window strategy (⑤, ★). As shown in Tab. 3, we further vary the window width $\ell$ and observe an optimum at $\ell=23$: larger windows weaken the locality and boundary precision, whereas smaller windows ($\ell < 23$) lack sufficient context and also degrade performance. In addition, we ablate the number of local windows. Using too few windows provides insufficient supervision, leading to worse results. Once the number of windows exceeds 100, the performance plateaus and remains comparable across settings, indicating that additional windows bring little benefit.

## 5 CONCLUSION

We presented **PatchRefiner V2**, an enhanced and efficient framework for high-resolution monocular metric depth estimation. Building on the strengths of the original PatchRefiner, PRV2 introduces a lightweight refiner branch, dramatically improving inference speed and reducing model size. With the novel Coarse-to-Fine (C2F) module and Noisy Pretraining strategy, our framework successfully mitigates the challenges posed by noisy features and the lack of pre-training of the refiner branch. Furthermore, we introduced the local Scale-and-Shift Invariant Gradient Matching (local SSIGM) loss to enhance boundary accuracy and improve synthetic-to-real transfer. Our framework significantly outperforms previous methods on the UnrealStereo4K dataset, achieving up to 9.2x fewer parameters and 10.7x faster inference. PRV2 also demonstrates considerable improvements in depth boundary delineation on real-world datasets.

## 6 ACKNOWLEDGEMENTS

The research reported in this publication was supported by funding from King Abdullah University of Science and Technology (KAUST) – Center of Excellence for Generative AI, under award number 5940 and a gift from Google.

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

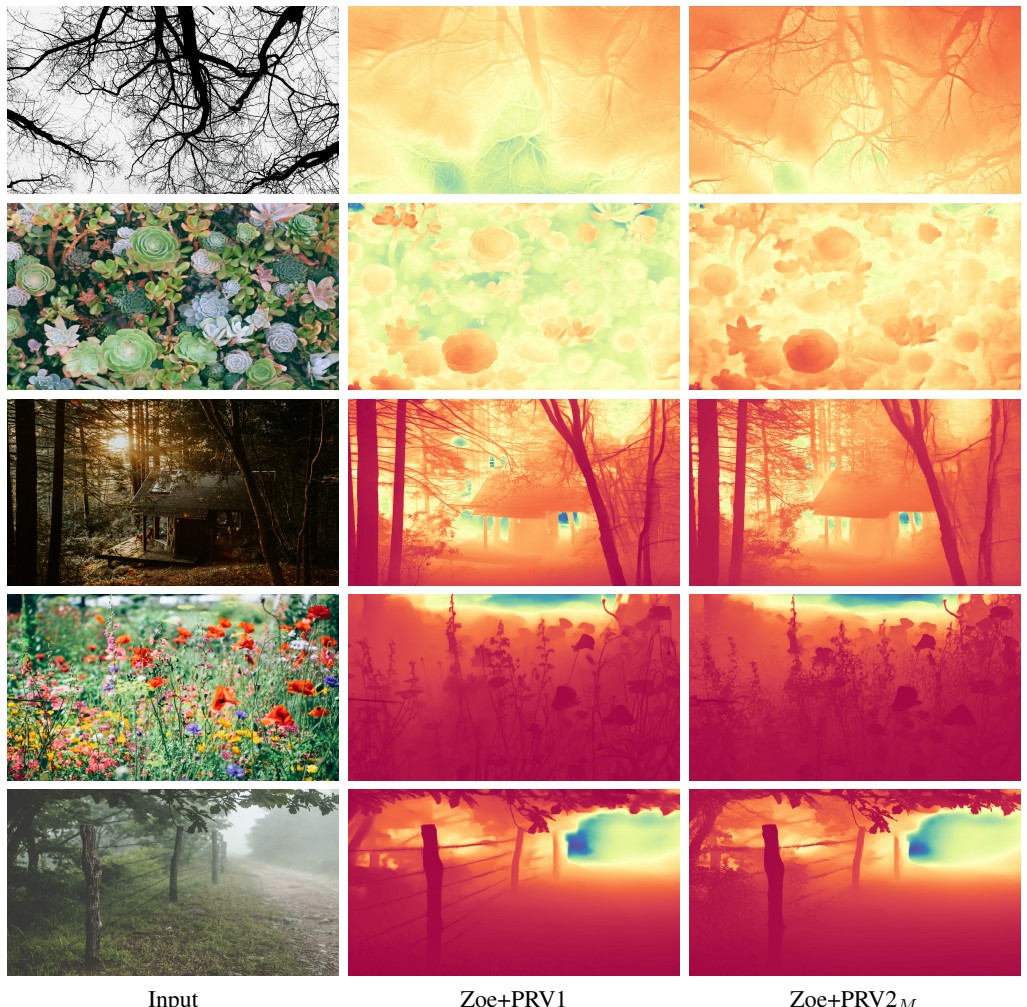

| Input | Zoe+PRV1 | Zoe+PRV2$_M$ |

Figure 5: **Qualitative Comparison with PRV1.** Images are from the internet. Though our PRV2$_M$ is 7.6x smaller and 4.5x faster than RRV1, it achieves satisfactory and comparable results. Zoom in to better perceive details near boundaries.

## A  DATASET

**UnrealStereo4K (Synthetic, 4K):** The UnrealStereo4K dataset Tosi et al. (2021) consists of synthetic stereo images with a resolution of 2160×3840 pixels, each paired with precise, boundary-complete pixel-wise ground truth. Images with labeling inaccuracies are excluded based on the Structural Similarity Index (SSIM) Wang et al. (2004), a process adapted from Li et al. (2024a;b). Ground truth depth maps are computed from the provided disparity maps using specific camera parameters. Consistent with the splits suggested in Tosi et al. (2021); Li et al. (2024a;b), the experiments utilize a patch size of 540×960 pixels for fair comparison.

**CityScapes (Real):** The CityScapes dataset Cordts et al. (2016) provides a diverse collection of urban scene images, segmentation masks, and disparity maps at a resolution of 1024×2048 pixels. This dataset surpasses many in its domain in terms of image density, volume, and resolution Silberman et al. (2012); Song et al. (2015); Schops et al. (2017); Scharstein et al. (2014). For our experiments, we use a standard patch size of 256×512 pixels, primarily focusing on this dataset for testing our models following Li et al. (2024b).

Table 4: **Ablation study of PRV2$_E$ on UnrealStereo4K.** F2C and C2F denote the fine-to-coarse and coarse-to-fine module in the bi-directional fusion module, respectively. E2E and NP are short for end-to-end training and noisy pretraining. Time: average inference time of the refiner branch for one image.

| | F2C | E2E | C2F | NP | Method | RMSE | #param. | T(s) |
|---|---|---|---|---|---|---|---|---|
| | | | | | Coarse Baseline | 1.289 | - | - |
| ① | ✓ | | | | | 1.118 | **51.7M** | **0.29s** |
| ② | ✓ | | | | | 1.185 | 95.6M | 0.47s |
| ③ | ✓ | ✓ | | | | 1.100 | **51.7M** | **0.29s** |
| ④ | ✓ | ✓ | ✓ | | | 0.985 | 72.1M | 0.57s |
| ★ | ✓ | ✓ | ✓ | ✓ | | **0.947** | 72.1M | 0.57s |

Table 5: **Ablation study of PRV2$_C$ on UnrealStereo4K.**

| | F2C | E2E | C2F | NP | Method | RMSE | #param. | T(s) |
|---|---|---|---|---|---|---|---|---|
| | | | | | Coarse Baseline | 1.289 | - | - |
| ① | ✓ | | | | | 1.095 | **226.9M** | **0.38s** |
| ② | ✓ | | | | | 1.151 | 270.9M | 0.61s |
| ③ | ✓ | ✓ | | | | 1.089 | **226.9M** | **0.38s** |
| ④ | ✓ | ✓ | ✓ | | | 0.946 | 245.8M | 0.62s |
| ★ | ✓ | ✓ | ✓ | ✓ | | **0.883** | 245.8M | 0.62s |

## B  QUALITATIVE COMPARISON WITH PRV1

We present the qualitative comparison with PRV1 in Fig. 5. Though our PRV2$_M$ is 7.6x smaller and 4.5x faster than RRV1, it achieves satisfactory and comparable results.

## C  ABLATION STUDY

**Framework Design:** We present more ablation studies about framework design based on PRV2$_E$ and PRV2$_C$ as shown in Tab. 4 and Tab. 5, respectively. we start with a baseline framework (①) in which we only substitute the base depth model in the refiner branch in PR with the lightweight encoder. While the inference time and model size are drastically reduced, quality is also degraded. Simply adding more parameters to scale up the F2C cannot improve the performance, as shown in ②. Adopting an end-to-end training strategy can help improve the model performance (③). With the help of our proposed C2F that denoises the refiner features effectively, the model's RMSE is reduced while only introducing a satisfactory overhead (③, ④). When equipped with the NP (★), our model achieves the best performance.

**PRV2 with Stronger Base Model:** We fine-tuned DepthPro Bochkovskii et al. (2025) separately on UnrealStereo4K and CityScapes using the same training protocol as PRV2. Integrating our PRV2 refinement module (e.g., PRV2$_M$) on top of DepthPro as the coarse branch yields notable gains on both trained domain and zero-shot context (see Tab. 6), demonstrating that our approach is complementary to existing SOTA coarse models under a fair resource setting.

We believe our method is extensible toward improving its zero-shot capability. Specifically: 1) It can be scaled up using stronger coarse backbones, leading to better generalization ability. 2) It can be trained jointly across multiple real domains, benefiting from our lightweight design and the effective local SSIGM loss. We regard this as a promising path for future work toward high-resolution zero-shot depth estimation. (3) Tab. 6b also suggests that our method has the potential to improve the zero-shot ability of DepthPro with the same training data.

**Weaker Base Model:** We adopt DenseDepth Alhashim & Wonka (2018) as the coarse model and train our PRV2$_M$. As shown in Tab. 7, our framework can consistently boost the model performance for high-resolution depth estimation, indicating the effectiveness of PRV2 even based on a weaker coarse estimator.

Table 6: **PRV2 with DepthPro as Base Model.**

(a) Cityscapes.

|  | Metric RMSE↓ | Boundary F1↑ |
|---|---|---|
| DepthPro | 7.341 | 29.46 |
| DepthPro + Ours | **7.257** | **37.51** |

(b) UnrealStere4K and ETH3D.

| Method | u4k | | ETH3D | |
|---|---|---|---|---|
|  | RMSE↓ | SEE↓ | $\delta$ ↑ | AbsRel↓ |
| DepthPro | 1.285 | 0.872 | 93.61 | 0.086 |
| DepthPro + PRV2$_M$ | **0.824** | **0.692** | **94.12** | **0.077** |

Table 7: **Framework Performance with a Weaker Base Model on UnrealStereo4K.** Our framework can consistently boost the model performance for high-resolution depth estimation.

| Method | RMSE↓ | SEE↓ |
|---|---|---|
| DenseDepth Alhashim & Wonka (2018) | 2.552 | 1.842 |
| DenseDepth + PRV2$_M$ | 1.898 | 1.436 |

Table 8: **NP *v.s.,* Metric3D Pretrained Weights on UnrealStereo4K.** It indicates the effectiveness of our NP strategy.

| Method | RMSE↓ | SEE↓ |
|---|---|---|
| Metric3D Pretrained Hu et al. (2024) | 0.931 | 0.798 |
| ours (with NP) | 0.883 | 0.787 |

**NP *v.s.,* Other Depth Pretrained Weights:** In this experiment, we adopt the Metric3D Hu et al. (2024) pretrained ConvNext Liu et al. (2022) as the refiner encoder. As shown in Tab. 8, our NP demonstrates its effectiveness with a 5.1% lower RMSE. Note that the Metric3D pretrained ConvNext is able to provide satisfactory depth-related features. Hence, such a discrepancy may indicate the importance of including the F2C and C2F modules in the pretraining stage, which is a core part of our NP strategy.

# D  LLM USAGE

We use ChatGPT to polish the paper writing.

