# OpenReview forum: "PatchRefiner V2: Fast and Lightweight Real-Domain High-Resolution Metric Depth Estimation"
_ICLR.cc/2026/Conference — ICLR 2026 Poster_

### Official Review · Reviewer_pdNs · 2025-10-28

**Soundness:** 3
**Presentation:** 3
**Contribution:** 4
**Rating:** 8
**Confidence:** 5

**Summary:**

This paper extends PatchRefiner (ECCV 2024) for high-resolution monocular metric depth estimation. PatchRefiner V2 replaces the heavy refinement encoder with a lightweight alternative to significantly reduce runtime and memory usage during both training and inference. To compensate for the reduced capacity, the method introduces a coarse-to-fine module that injects coarse-level features to guide and denoise refinement features, along with a noisy pretraining strategy to improve the refiner's robustness. The paper also adopts an improved local-window SSIGM loss to enhance synthetic-to-real transfer. Experiments on UnrealStereo4K and Cityscapes demonstrate improved accuracy and efficiency compared to the original PatchRefiner.

**Strengths:**

- The proposed PatchRefiner V2 achieves impressive improvements in runtime and memory efficiency over the original PatchRefiner, while also achieving equal or better accuracy. This makes high-resolution monocular depth estimation more feasible for practical use.

- The coarse-to-fine module is well-motivated and effectively compensates for the reduced capacity of the lightweight refiner. The guided denoising design is clearly presented, and its contribution is supported by thorough ablation studies.

- The noisy pretraining strategy is simple yet empirically effective. Pretraining the refinement branch with randomized coarse features leads to a more robust model without requiring additional external data or complex setups.

- The local SSIGM loss provides more precise supervision. Enforcing scale-and-shift consistency at a local spatial level results in sharper depth boundaries and avoids introducing scale error.

- The paper is well-written and easy to follow, with clear figures and architectural diagrams that effectively explain the design choices.

- The visual annotations (e.g., the snails and lightning bolts in Fig. 1) clearly and intuitively highlight the performance and efficiency differences. The visualization of features in Fig. 2 clearly presents the motivation and effectiveness of the fusion model. These figures improved readability and made the narrative smoother and more engaging.

**Weaknesses:**

- Discussion of Related Work Could Be Expanded.
The motivation and effectiveness of the proposed coarse-to-fine module are clearly presented, and the design is well-justified. However, there are existing two-branch feature fusion strategies in related areas (e.g., [1] and [2]). While these works focus on different tasks and modalities, a brief discussion comparing the design philosophy or fusion flow direction could further clarify the novelty of the proposed C2F module and situate the contribution more explicitly in the broader literature.

[1] Bi-SSC: Geometric-Semantic Bidirectional Fusion for 3D Scene Completion

[2] FFB6D: Full Flow Bidirectional Fusion for 6D Pose Estimation

- Clarification of PRV2’s Advantage Over High-Resolution Backbone Models.
The improvement of PRV2 over DepthPro is quite substantial, which strongly supports the value of the refinement design. Since DepthPro is already a high-resolution metric depth model, it would be helpful for the paper to provide a bit more insight into why PRV2 achieves such notable gains when refining DepthPro outputs.

- While PRV2 is not intended to be a general-purpose “zero-shot” depth refiner, a short discussion of the expected generalization behavior could help guide future follow-up work aiming toward more foundational refinement pipelines.


- There is a small typo in Table 2: in the caption, “GM and wins.” can be removed for clarity.

- Since the updated local-window SSIGM loss is one of the key improvements, adding a brief pseudo-code snippet in the supplement would make re-implementation easier. This would improve the usability.

**Questions:**

Please refer to the weaknesses.

---

> ### Author Response · Authors · 2025-11-20
> **Reply to R4**
>
> # Reply to R4
>
> ## R4:Q1
>
> Thank you for the suggestion. We will expand the related work section to briefly discuss the suggested references. First, Bi-SSC and FFB6D target different tasks (3D completion and 6D pose estimation). In addition, their bidirectional fusion serves a different purpose from ours: PRV2’s C2F module is specifically designed to **improve lightweight encoder features using coarse depth guidance**, addressing the depth-alignment issue introduced by removing heavy pretrained backbones. We refer to R3:Q6 for more discussions and will clarify this point in the revision.
>
>
> ## R4:Q2
>
> We appreciate this point. DepthPro is limited to a fixed maximum inference resolution (1536×1536), while PRV2 performs patchwise refinement at **native resolution** (3840x2160), containing 3.51 times more pixels than DepthPro's resolution. This enables substantial boundary sharpening. We will add a short explanation of this in the paper.
>
>
> ## R4:Q3
>
> Thank you for the suggestion. We refer to R2:Q2 and R3:Q3 for more experimental results that demonstrate the generalization capability of our method.
>
> At present, the increase of generalization is constrained by the scale and diversity of available high-resolution depth datasets. We expect that, with access to larger and more diverse training data, the lightweight refiner could generalize better and potentially serve as a strong universal depth-refinement module in future work.
>
>
> ## R4:Q4
>
> Thank you for catching these. We have removed the “GM and wins.” fragment from Table 2.
>
> ## R4:Q5
>
> Here we present a pseudo-code snippet for the local SSIGM computation:
>
> Algorithm: Local SSIGM Loss
>
> Hyperparams: window size ℓ, number of windows N
>
> 1. Sample N valid window centers {(xk, yk)}.
>
> 2. For each center k:
>
>       Ω ← ℓ × ℓ window around (xk, yk)
>
>       Local scale–shift alignment (least squares):
>
>       (s, t) ← alignment(pred, pseudo)
>
>       aligned prediction   ← s · pred + t
>
>       Loss calculation
>
>       r = pseudo - aligned prediction
>
>       L_k ← gm_loss(r)
>
> 3. Return the average of {L_k}.

---

> > ### Comment · Reviewer_pdNs · 2025-11-22
> >
> > Thanks for the positive feedback which has addressed my concerns. I vote for acceptance.

---

### Official Review · Reviewer_JkXG · 2025-10-30

**Soundness:** 3
**Presentation:** 3
**Contribution:** 3
**Rating:** 6
**Confidence:** 5

**Summary:**

This paper presents PatchRefiner V2 (PRV2), an enhanced framework for high-resolution monocular depth estimation that aims to address the computational inefficiencies of its predecessor, PatchRefiner (PRV1). The core contributions are a lightweight refiner branch, a novel Coarse-to-Fine (C2F) module with Guided Denoising Units (GDUs), a Noisy Pretraining (NP) strategy, and a local Scale-and-Shift Invariant Gradient Matching (local SSIGM) loss. The paper is well-structured, the problem is clearly motivated, and the experimental evaluation is comprehensive.

**Strengths:**

1. The paper effectively identifies the critical bottlenecks of PRV1—high inference time, large memory footprint, and the inability for end-to-end training—due to using a heavyweight base model for patch-level refinement. The motivation for replacing it with a lightweight encoder is well-justified and addresses a practical need for real-world applications.
2.The idea of using coarse features to "denoise" the features from a lightweight encoder is intuitive and effective. The GDU mechanism is clearly explained and visualized showing a tangible improvement in feature quality.
3. Noisy Pretraining is a simple yet clever strategy to force the refiner branch to learn robust, depth-relevant features from the high-resolution input itself. The ablation studies strongly validate its importance.
4. Extending the SSI loss to the gradient domain and applying it within local windows is a thoughtful approach to improve boundary accuracy without compromising global scale. The significant improvement in the boundary F1 score on CityScapes is a key result.
5. The paper provides thorough quantitative and qualitative evidence. The results on UnrealStereo4K are impressive, demonstrating that PRV2 can achieve state-of-the-art or comparable accuracy with a massive reduction in parameters (up to 9.2x) and inference time (up to 10.7x faster). The ablation studies are systematic and clearly demonstrate the contribution of each proposed component (C2F, NP, E2E training, local SSIGM).
6. The inclusion of experiments on a real-world dataset (CityScapes) and the analysis of boundary quality are highly valuable and demonstrate the method's practical utility.
7. The method is described in sufficient detail, with clear diagrams and mathematical formulations for the GDU and local SSIGM loss. The implementation details provided in Section 4.2 are adequate for reproduction.

**Weaknesses:**

1The GDU is a central component, but the ablation only compares it to one alternative. A more detailed analysis, for instance, comparing the proposed sigmoid-based gating to an additive fusion or an attention-based mechanism, would provide deeper insights into why the current design is optimal.
2.  While the overall framework is much faster, the specific computational cost introduced by the C2F module and the local SSIGM loss (during training) is not discussed. A brief note on their relative overhead would be useful for readers considering implementation.
3. The experiments are focused on UnrealStereo4K and CityScapes. While the results on CityScapes show good synthetic-to-real transfer, a brief zero-shot evaluation on other standard depth benchmarks (e.g., KITTI, NYUv2) would more strongly demonstrate the generalizability and robustness of the learned representations, especially given the use of the local SSIGM loss.
4. The field of efficient high-resolution vision is rapidly evolving. A discussion of how PRV2 compares to other contemporary lightweight or patch-based refinement approaches (beyond PRV1 and PatchFusion) would better situate its contributions within the current research landscape.
5. Some arxiv papers are actually published on important conferences, and please cite the published information, not arxiv.

**Questions:**

While the term noisy features is used to describe the output of the lightweight encoder, a more precise characterization would strengthen the argument. Is this noise in the traditional sense (random, high-frequency artifacts), or is it a lack of depth-specific semantic structure? A brief quantitative analysis (e.g., using feature similarity metrics) could complement the visual evidence.

---

> ### Author Response · Authors · 2025-11-20
> **Reply to Q1-Q4 from R3**
>
> ## R3:Q1
>
> Following the reviewer’s recommendation, we conducted additional ablations comparing the proposed GDU with several widely used fusion mechanisms, including (i) concatenation followed by convolution (the default PatchRefiner fusion, already reported in Tab. 2 ⑦), (ii) additive fusion, and (iii) a self-attention fusion block [1]. The results on UnrealStereo4K (MobileNet refiner) are:
>
> |    |RMSE|
> |----|----|
> |Concatenation + Conv  (used in PRV1)    |1.202    |
> |Additive Fusion | 1.210    |
> |Self-Attention Fusion [1]    | OOM    |
> |Ours  | 1.041    |
>
> Our GDU consistently outperforms these alternatives, which supports our design choice. Specifically, the performance of the additive fusion mechanism is on par with the baseline, as it lacks the ability to selectively suppress noisy refiner features. The self‑attention fusion resulted in OOM on 80GB NVIDIA A100 due to the quadratic memory cost on high‑resolution feature maps, which are essential for PRV2’s multi‑resolution refinement.
>
> ## R3:Q2
>
> We appreciate the opportunity to clarify this point.
>
> For the C2F module, Tab. 2 (left) already provides the number of parameters and inference time of different fusion designs. The basic PatchRefiner fusion module uses 27.5M parameters with 0.08s inference time (row ①). Simply scaling up this baseline to 70.2M parameters increases the inference time to 0.38s but yields no performance benefit because the underlying noisy-feature issue is not addressed (row ②). In contrast, incorporating our C2F module results in a 47.0M-parameter fusion block with 0.32s inference time but significantly improves RMSE (row ④). This demonstrates that the additional overhead introduced by C2F is modest relative to the performance benefit.
>
> For the local SSIGM loss, using the same training setup, one epoch of training with the baseline SSI loss takes 1907s, while training with SSIGM takes 1926s, corresponding to approximately +0.99% additional time per epoch.
>
>
> ## R3:Q3
>
> Thank you for the suggestion. To benchmark cross-dataset generalization, we conducted a zero-shot evaluation on KITTI, using PRV1 and PRV2 models trained solely on UnrealStereo4K and Cityscapes under the synthetic-to-real protocol, with no fine-tuning on KITTI. The results are:
>
> |  |Scale RMSE |Boundary SEE |# of params |T(s)
> |---|---|---|---|---|
> |ZoeDepth + PRV1             |3.379 |1.628|369.0M |1.45s |
> |ZoeDepth + PRV2$_E$ |3.422 |1.593|72.1M  |0.57s |
>
> PRV2 achieves comparable scale RMSE and slightly better boundary SEE than PRV1, while being 5.1× smaller and 2.5× faster. This indicates the effectiveness of our PRV2.
>
> ## R3:Q4
>
>
> Thank you for this observation. We have expanded the discussion to better situate PRV2 within the context of recent work.
>
> A recent patch-based refinement approach [2] introduces Grouped Patch Consistency Training and Bias-Free Masking to improve patch consistency and mitigate dataset-specific biases. Their contribution focuses on consistency learning and *relative depth*, whereas PRV2 centers on efficient lightweight architecture design and *metric depth*. Consistency learning and efficient lightweight architecture design are orthogonal and complementary, whereas *relative depth* and *metric depth* are two different challenges.
>
> Additionally, _DepthPro_ represents another strong high-resolution depth framework. As shown in Table 6 in the supplementary material, integrating PRV2 into DepthPro consistently improves the performance. The difference in inference resolution is noteworthy: DepthPro is constrained to ~1536×1536, whereas PRV2’s patch-based design can handle arbitrarily high resolutions. This reinforces the complementary strengths of our approach within the current landscape of high-resolution depth estimation.

---

> ### Author Response · Authors · 2025-11-20
> **Reply to Q5-Q6 from R3**
>
> ## R3:Q5
>
> Thank you for pointing this out.  We have updated the references to cite the published conference versions where appropriate.
>
>
> ## R3:Q6
>
>
> Thank you for raising this important question. In our context, we use the term "noisy features" not to refer to traditional random noise, but rather use it to describe the **less informative feature representations** produced by lightweight encoders. These features are generally less helpful for high-resolution depth refinement, especially around boundaries, due to the lack of strong depth-specific semantic cues. We will clarify this point in our revised paper.
>
> To quantitatively validate the denoising effect of our Coarse-to-Fine (C2F) module, we analyze the alignment between refiner feature boundaries and ground-truth depth boundaries. Specifically, we extract gradient-based boundary masks from the refiner features (with and without C2F), and compare them to the ground-truth depth boundary using standard boundary detection metrics:
>
> Feature Type |Precision |Recall |F1 Score |IoU |
> |---|---|---|---|---|
> Without C2F |0.290 |0.145 |0.193 |0.109
> With C2F (Ours) |0.455 |0.227 |0.303 |0.182
>
> These results demonstrate that the features refined with C2F yield sharper and more geometrically aligned boundaries, making them more suitable for high-resolution depth refinement. We will include this quantitative analysis and clarify the use of the term "noisy features" in the revised paper.
>
> [1]: Wang X, Girshick R, Gupta A, et al. Non-local neural networks[C]//Proceedings of the IEEE conference on computer vision and pattern recognition. 2018: 7794-7803.
>
> [2]: Kwon B, Kim M. One Look is Enough: Seamless Patchwise Refinement for Zero-Shot Monocular Depth Estimation on High-Resolution Images[C]//Proceedings of the IEEE/CVF International Conference on Computer Vision. 2025: 8077-8087.

---

### Official Review · Reviewer_JEWx · 2025-10-31

**Soundness:** 3
**Presentation:** 3
**Contribution:** 3
**Rating:** 6
**Confidence:** 4

**Summary:**

This paper presents a refined version of PatchRefiner, aiming to accelerate inference by replacing the original patch refiner with a lightweight network. The proposed approach incorporates a coarse-to-fine module, a guided denoising unit, and a noisy pre-training strategy. With these enhancements, PatchRefiner V2 achieves substantial speed improvements while delivering superior performance compared to its predecessor.

**Strengths:**

- The paper is well written and clearly organized.
- The proposed improvements to PatchRefiner demonstrate both significant acceleration in inference speed and enhanced performance.

**Weaknesses:**

1. At Line 377, the authors state that the Cityscapes dataset is used for synthetic-to-real transfer evaluation. However, quantitative comparisons with other methods are missing in both the main text and the supplementary material, which limits the completeness of the evaluation.
2. Only quantitative results on the in-domain UnrealStereo4K dataset are reported. Considering that the base model, ZoeDepth, is a generalizable depth estimator, it would be valuable to include experiments under cross-dataset settings to provide a more comprehensive assessment of the proposed method.
3. At Line 365, the authors claim that local SSIGM performs better than matching gradients over the entire map. However, as shown in Table 3, the variant with zero windows does not exhibit significant performance degradation compared to local variants, and the influence of window size and number of windows on performance appears minimal. This observation reduces the perceived effectiveness of the proposed local SSIGM loss.

**Questions:**

1. Regarding the noisy pre-training strategy, it would be helpful to clarify whether the type of noise used during training influences the final performance. For instance, how would the results change if Gaussian noise were replaced with uniform noise?

---

> ### Author Response · Authors · 2025-11-20
> **Reply to R2**
>
> ## R2:Q1
>
> Thank you for pointing this out. The quantitative comparison on Cityscapes _is already included_ in **Table 2 (Right)** of the main paper, where we compare:
>
> -   **Coarse baseline (ZoeDepth)**
>
> -   **ZoeDepth + PRV2 architecture + PRV1 DSD Loss**
>
> -   **ZoeDepth + PRV2 architecture + PRV2 DSD Loss**
>
> We agree that the current caption does not explicitly highlight which row corresponds to **PRV1**, which may have caused confusion.  In the revision, we updated the table caption to clarify this comparison.
>
> In Table 2 (Right), we mainly aim to highlight the effectiveness of the newly proposed DSD Loss. Below, we present a more comprehensive and concise comparison, including the PRV1 architecture.
>
> |     |Scale RMSE    |Boundary F1   |#param |T(s)                      |
> |----|----|----|----|----|
> |ZoeDepth                    |9.097    |19.15   |
> |ZoeDepth + PRV1 arch. + PRV1 DSD Loss      |8.313    |30.02    |369.0M |1.45s
> |ZoeDepth + PRV1 arch. + PRV2 DSD Loss      |8.293    |37.05    |369.0M |1.45s
> |ZoeDepth + PRV2 arch. + PRV1 DSD Loss      |8.533    |29.22    |72.1M |0.57s
> |ZoeDepth + PRV2 arch. + PRV2 DSD Loss      |8.527    |36.54    |72.1M |0.57s
>
> These results demonstrate the effectiveness of our PRV2.
>
>
> ## R2:Q2
>
> Thank you for this suggestion. Following the reviewer’s recommendation, we additionally evaluate our method in a zero-shot setting on ETH3D, a high-resolution real-domain dataset. We use the official evaluation protocol [1] and apply our models without any fine-tuning (Results using DepthPro as the coarse model are already included in Supplementary Table 6(b)).
>
> DepthPro Results: When adopting DepthPro as the coarse model, PRV1 cannot be applied due to memory limitations: its dual-branch design requires the refiner to share the same heavy architecture as the coarse branch, causing out-of-memory issues and forcing the patch resolution to be restricted to 1536×1536. In contrast, PRV2 uses a lightweight refiner and therefore fits comfortably within memory, enabling flexible high-resolution inference and consistently improving DepthPro’s performance even in a zero-shot setting.
>
> ZoeDepth Results: The zero-shot results using ZoeDepth as the coarse model are summarized below.
>
> |  |Delta$_1$ |AbsRel |# of params |T(s)
> |---|---|---|---|---|
> |ZoeDepth            |89.8 |0.089 |-      | -    |
> |ZoeDepth + PRV1     |94.7 |0.076 |369.0M |1.45s |
> |ZoeDepth + PRV2$_E$ |94.4 |0.078 |72.1M  |0.57s |
>
> As shown in the table, PRV2 consistently improves zero-shot performance over the base estimator, indicating that our refinement module can transfer effectively to new domains without additional training. Compared to PRV1, PRV2 achieves comparable results, but it’s **2.5x faster and 5.1x smaller** than PRV1.
>
> ## R2:Q3
>
> Thank you for raising this point. We clarify that **local SSIGM contains two orthogonal contributions** over PRV1's DSD Loss:
>
> ### (1) Gradient matching term (SSIGM)
>
> This is where the main gain comes from.  Compared to PRV1’s pseudo-labeling (ranking and SSI), introducing gradient matching already improves boundary F1 from **29.22** to **33.47** (**+14.5%**, Tab. 2b rows ③ and ④). This confirms that matching gradients in the aligned space contributes most to the improvement.
>
> ### (2) Local-window formulation
>
> Used with the gradient matching term and NP, this local-window strategy provides additional refinement, improving boundary F1 from **35.32** to **36.54** (**+3.5%**,  Tab. 2b rows ⑤ and ★)
>
> Here, we conduct one experiment, directly adopting our local-window strategy on top of the PRV1's SSI loss, without the gradient matching term and NP:
>
> |     |Scale RMSE    |Boundary F1   |
> |----|----|----|
> |with SSI                  |8.533 |29.22   |
> |with SSI + local windows  |8.556 |35.04   |
>
> This experiment better isolates the contribution of our local-window strategy, which leads to a +19.9% boundary F1 improvement over the SSI loss.
>
>
> ## R2:Q4
>
> Thank you for the question. We experimented with **Gaussian noise (N(0,1))**, **uniform noise (U[-1,1])**, and **constant values (0)** for the coarse-branch pseudo-features.
>
> We observe that Gaussian and Uniform noise perform nearly identically.  This indicates that NP does not depend on the specific noise distribution; what matters is the _stochastic variation_ that forces the refiner to ignore coarse features and extract depth-relevant structure directly from image patches.
>
> However, constant-value “noise” (e.g., all zeros) leads to a failure of NP. The model overfits to the constant tensor and becomes overly reliant on this artificial signal during pretraining.  When switching to real coarse features during fine-tuning, the mismatch significantly harms performance.
>
> [1]: Kwon B, Kim M. One Look is Enough: Seamless Patchwise Refinement for Zero-Shot Monocular Depth Estimation on High-Resolution Images[C]//Proceedings of the IEEE/CVF International Conference on Computer Vision. 2025: 8077-8087.

---

> > ### Comment · Reviewer_JEWx · 2025-11-24
> >
> > Thank the author for their detailed feedback. Most of my concerns are addressed after reading the response. Regarding the first question, my suggestion is to add comparison with existing methods (not only the ablation study) on CityScapes. In summary, I think this paper is a good extension for its previous version and I would like to raise my rating to 8.

---

> > > ### Author Response · Authors · 2025-11-24
> > > **Reply to R2**
> > >
> > > We thank the reviewer for the thoughtful feedback and for raising the score. We appreciate your suggestion regarding broader comparisons on CityScapes and will include additional results with existing methods in the final version to strengthen the evaluation.

---

### Official Review · Reviewer_k7TN · 2025-11-04

**Soundness:** 3
**Presentation:** 3
**Contribution:** 2
**Rating:** 4
**Confidence:** 2

**Summary:**

This paper proposed PatchRefiner V2(PRV2) for real-domain high-resolution metric depth estimation. PRV2 replaces a heavyweight refiner  with lightweight encoder and adds a coarse-to-fine (C2F) block with a guided denoising unit. The paper also present noisy pretraining. Experiments on UnrealStreao4K  (synthetic) and Cityscapes (real) report the efficiency of PRV2.

**Strengths:**

1. Experimental results
On UnrealStereo4K, PRV2 delivers state-of-the-art accuracy while using fewer parameters and achieving faster inference than strong baselines.

2. Simple design & recipe
A modular architecture and a minimal noisy-pretraining scheme make the method easy to reproduce and extend into existing pipelines.

**Weaknesses:**

1. Benchmark coverage is narrow (mainly UnrealStereo4K)
The main SOTA claims are substantiated primarily on one synthetic dataset, while the real-domain evidence is limited to Cityscapes.

2. Incomplete comparators for 2024–2025 SOTA
UnrealStereo4K comparisons largely focus on ZoeDepth/ZoeDepth+PF/ZoeDepth+PRV1. Please add or discuss comparisons (or a justified protocol mismatch) against strong recent depth estimation methods (e.g., Marigold, SharpDepth, ...) matched resolution/compute, and clarify where a fair comparison is infeasible.

3. Ablations split across datasets create interpretation friction
Core architectural ablations (C2F/NP) are on UnrealStereo4K, while loss/boundary analyses are on Cityscapes, which makes it hard to see how each module contributes on the same data. Please add a unified ablation table on one dataset (preferably a real set) so readers can read row-wise improvements coherently

4. Contribution novelty leans engineering rather than conceptual
GDU-style gating and the NP strategy are practical and effective, but feel incremental relative to prior guided fusion paradigms

5. Timing definition under-reports full pipeline cost
The paper defines T as the refiner-branch time per image; please also report end-to-end wall-clock (coarse + all patch refinements) and memory for a fair, reproducible comparison for follow-up research

**Questions:**

No questions

---

> ### Author Response · Authors · 2025-11-20
> **Reply to Q1-Q3 from R1**
>
> ## R1:Q1
>
> Thank you for pointing this out. We agree that additional benchmarks will strengthen the contribution.
>
> We present **one additional synthetic benchmark** and **two additional real-domain benchmarks** using the same evaluation setup as PatchFusion and PatchRefiner for a fair comparison.
>
> ### **(A) Synthetic Benchmark: MVS-Synth [1]**
>
> We follow the official PatchFusion protocol (same patch size, same training split, same evaluation metrics).
>
> PRV2 consistently improves over PRV1 on this benchmark. Note that our PRV2 achieves similar performance as PRV1, whereas it's **2.5x faster and 5.1x smaller** than RPV1.
>
> |    |delta$_{1}$   |RMSE    |# of params | T(s)
> |--|--|--|--|--|
> |ZoeDepth                    |93.978    |1.2676    | - | - |
> |ZoeDepth + PatchFusion      |95.991    |0.9213    |432.7M |3.44s |
> |ZoeDepth + PRV1             |96.703    |0.8049    |369.0M |1.45s |
> |ZoeDepth + PRV2$_E$ |96.797    |0.8032    |72.1M |0.57s |
>
> ### **(B) Real-Domain Benchmarks: KITTI & ScanNet**
>
> Following PatchRefiner’s exact evaluation protocol (scale RMSE and boundary SEE), we compare it against the main baseline, PRV1, using ZoeDepth as the coarse model.
>
> |  |KITTI |KITTI |ScanNet |ScanNet |
> |---|---|---|---|---|
> |  |Scale RMSE |Boundary SEE |Scale RMSE |Boundary SEE |
> |ZoeDepth                    |2.344 |0.906 |0.268 |0.215|
> |ZoeDepth + PRV1             |2.155 (+8.06%) |0.863 (+4.75%) |0.268 (+0.00%) |0.216 (-0.47%)|
> |ZoeDepth + PRV2$_E$ |2.154 (+8.11%) |0.849 (+6.29%) |0.268 (+0.00%) |0.207 (+3.72%)|
>
> Across all three added benchmarks, **PRV2 consistently improves over PRV1**, demonstrating that the method generalizes beyond UnrealStereo4K.
>
>
> ## R1:Q2
>
> Thank you for the suggestion. Following your suggestion, below, we discuss comparisons or a justified protocol mismatch against strong recent depth estimation methods, including **Marigold (CVPR 2024), GenPercept (ICLR 2025) [2], MoGe (CVPR 2025), SharpDepth (CVPR 2025) [3], DepthPro (ICLR 2025)**. Since we show good results on several base models (ZoeDepth, DepthAnythingV2, and DepthPro), we believe it is reasonable to expect that our contributions will be beneficial to future backbones.
>
> ### **1. Marigold, GenPercept, MoGe**
>
> (A) presents a category of depth estimation methods focusing on the *relative* depth estimation. When evaluating this kind of method, ground-truth depth is required for scale and shift alignment. However, our work focuses on metric depth estimation, where no ground-truth alignment is required for evaluation. This is a protocol mismatch in the evaluation, and it's unfair to include them in our benchmark. The other methods are not able to provide metric depth.
>
>
> ### **2. SharpDepth**
>
> SharpDepth also reports results on UnrealStereo4K. As presented in their Arxiv-version paper (https://arxiv.org/pdf/2411.18229), SharpDepth achieves a 0.47 Abs Rel on UnrealStereo4K, which is significantly worse than our PRV2's 0.034. This gap is introduced by the different training and evaluation protocols. In SharpDepth, the authors train their model using a combination of real-world datasets and evaluate it on UnrealStereo4K in a *zero-shot* manner. Though it's unfair to include this method in our benchmark, we have added discussions about this method in our paper.
>
>
> ### **3. DepthPro**
>
> We already included DepthPro comparisons in the supplementary (supp. Table 6).  We fine-tuned DepthPro using its official weights on UnrealStereo4K and CityScapes for a fair comparison. Despite being designed for “HR depth,” DepthPro’s fixed input size (1536×1536) limits performance on 4K scenes. Integrating PRV2 as a refinement module further improves DepthPro on UnrealStereo4K (supp. Table 6 right). Our framework can also improve the performance of the fine-tuned DepthPro model on the ETH3D dataset (supp. Table 6 right). For the real-domain Cityscapes benchmark (supp. Table 6 left), our framework consistently boosts the performance of DepthPro by a large margin.
>
>
> ## R1:Q3
>
> Thank you for pointing this out. **We have updated more ablation study results on Cityscapes in Table 2 right**. This row-by-row ablation now enables readers to understand the contribution of each component without needing to switch tables. We will also keep the UnrealStereo4K ablation for completeness.
>
> We also want to clarify *why* some ablations are performed on synthetic data and others on real data. As discussed in PatchRefiner (Sec. 3.2), **training high-resolution depth models directly on real-domain datasets is fundamentally limited** due to the lack of real high-resolution datasets and the missing boundary regions in ground-truth depth (even when the datasets claim a higher resolution, too many critical depth values are missing).
>
> Even under these real-data constraints, the unified ablations on Cityscapes (Table 2b) still show **consistent improvements** from each module of PRV2, further demonstrating the effectiveness of our design.

---

> ### Author Response · Authors · 2025-11-20
> **Reply to Q4-Q5 from R1**
>
> ## R1:Q4
>
> We appreciate this feedback. We clarify the conceptual contributions more explicitly:
>
> (A) GDU solves a new problem introduced by lightweight refinement. Prior fusion modules assume backbone features are depth-aligned. After replacing the refiner backbone with a lightweight encoder, we observed a new issue not discussed in prior work: refiner features become noisy and misaligned (Figure 2).
>
> The GDU is not simply a “gating” variant. It is specifically designed to inject coarse-depth features as denoising guidance, suppress noisy features from lightweight encoders, recover boundary precision, and enable the bidirectional fusion to function. We refer to R3:Q6 for more discussions and will clarify this point in the revision.
>
> Replacing GDU with prior fusion designs degrades performance significantly (Table 2, row ⑦), showing that PRV2 introduces an essential and novel fusion behavior for lightweight refiners. More experiments requested by Reviewer 3, comparing GDU with other fusion options, further demonstrate the effectiveness of our method.
>
> (B) For Noisy Pretraining (NP),  existing depth-estimation frameworks pretrain only the encoder. However, in PRV2 the encoder is <2% of the refiner parameters; the C2F/F2C/fusion modules (~98%) have never been pretrained in prior work.
>
> NP pretrains the entire refiner branch, including fusion layers, using randomly generated coarse features to avoid architectural coupling. It is designed to further unlock the potential and effectiveness of (A). Previous fusion methods ignore this point.
>
> ## R1:Q5
>
> Thank you for pointing this out. We report the efficiency comparison of the complete framework for a single input image here. For the shared ZoeDepth coarse branch, the inference time is ~0.24s.
>
> |    |T(s)   | Memory  | Flops |
> |----|----|----|----|
> |PatchFuison     |3.70   | 33.636G |28.748T
> |PRV1            |1.69   | 11.385G |14.418T
> |PRV2$_M$|0.56   | 5.612G  |7.542T
> |PRV2$_E$|0.78   | 6.949G  |7.666T
> |PRV2$_C$|0.85   | 8.268G  |9.95T
>
> [1] Huang P H, Matzen K, Kopf J, et al. Deepmvs: Learning multi-view stereopsis[C]//Proceedings of the IEEE conference on computer vision and pattern recognition. 2018: 2821-2830.
>
> [2] Xu G, Ge Y, Liu M, et al. What matters when repurposing diffusion models for general dense perception tasks?[J]. arXiv preprint arXiv:2403.06090, 2024.
>
> [3] Pham D H, Do T, Nguyen P, et al. Sharpdepth: Sharpening metric depth predictions using diffusion distillation[C]//Proceedings of the Computer Vision and Pattern Recognition Conference. 2025: 17060-17069.

---

### Author Response · Authors · 2025-11-20
**Authors' Reply to All Reviewers**

We thank the reviewers (R1 k7TN, R2 JEWx, R3 JkXG, R4 pdNs) for their feedback on our work. In this rebuttal, we provide clarifications, additional experimental results, and revisions addressing all raised concerns. Below, we respond to each reviewer point-by-point. Revisions made to the paper are highlighted in red in the updated manuscript.

---

### Author Response · Authors · 2025-12-01
**Official Comment by Authors**

Dear Reviewers and Area Chair,

We sincerely thank all reviewers (R1 k7TN, R2 JEWx, R3 JkXG, R4 pdNs) for their feedback and engagement with our submission. Below, we provide a summary of our paper and rebuttal.

### Reviewer Ratings

R2 (JEWx) raised their score from 6 → 8 after reading the rebuttal.

R4 (pdNs) maintained a strong score of 8, expressing full support for acceptance.

R1 (k7TN) gave a score of 4 but noted their low confidence of 2. The review expressed concerns about the evaluation and suggested additional related work as the main points of requested improvement.

R3 (JkXG) gave a score of 6, praising novelty and efficiency but suggesting more analysis on fusion design and generalization.

Overall, reviewers agree that PatchRefiner V2 (PRV2) presents a well-motivated and practical contribution toward efficient high-resolution metric depth estimation, improving over PRV1 in both accuracy and efficiency.

### Summary of Key Strengths Highlighted by Reviewers

Efficient High-Resolution Depth Estimation (R1, R2, R3, R4): PRV2 achieves 2.5× speedup and 5.1× fewer parameters compared to PRV1, with improved depth boundary accuracy.

Lightweight Architecture (R1, R2, R3, R4): The introduction of the Guided Denoising Unit (GDU) and Noisy Pretraining (NP) received positive feedback for their intuitive design and practical impact.

Real-Domain Performance (R3, R4): Reviewers noted improved depth boundaries on CityScapes and appreciated the effectiveness of the proposed local scale-and-shift invariant gradient matching loss (SSIGM) for real-domain transfer.

Clear Writing and Strong Visualizations (R2, R3, R4): The structure, figures, and clarity of presentation were commended across multiple reviews.

### Summary of Reviewer Concerns and Our Response

(1) Evaluation Scope (R1, R2, R3)

We expanded evaluation beyond UnrealStereo4K and CityScapes to include KITTI, ScanNet, ETH3D, and MVS-Synth. In total, we added three real benchmarks and one synthetic benchmark.
PRV2 consistently outperforms PRV1 across all settings.

(2) Comparisons with Recent Works (R1, R2, R3)

We discussed protocol mismatches (e.g., metric vs. relative depth) for cases where a comparison is not possible and added comparisons or contextual discussions for DepthPro, SharpDepth, Marigold, GenPercept, and others.

(3) Unified Ablation Table (R1) and Fusion Design Analysis (R3)

We added a row-wise unified ablation on Cityscapes to show the contribution of each module on real data.

We conducted new ablation experiments comparing GDU to additive fusion and self-attention fusion. GDU showed the best performance.

Self-attention fusion caused OOM due to high memory cost at large resolutions.

(4) Generalization (R2, R3, R4)

We conducted zero-shot evaluations on multiple datasets and showed consistent improvements over PRV1 and base models, validating PRV2’s transferability.

(5) Clarification on Loss Design (R2, R4)

We clarified the gradient matching vs. local windowing contributions in SSIGM and isolated their effects. Local windows improve boundary F1 by up to +19.9%.

(6) Qualitative Analysis of “Noisy Features” (R3)

We added quantitative evidence showing how C2F improves alignment with GT depth boundaries. F1 increases from 0.193 to 0.303.

### Conclusion

We believe the rebuttal has successfully addressed all the raised concerns. With strong support from R2 and R4, and the alignment of R3’s feedback with our added analysis, we hope the Area Chair and reviewers will find our submission significantly strengthened by considering feedback from the helpful reviews. PRV2 offers a practical and extensible framework for fast, lightweight, high-resolution depth estimation—a critical need for real-world applications in robotics, AR/VR, and beyond.

We sincerely appreciate your time and consideration.

Best regards,

The Authors

---

### Meta-Review · Area_Chair_LWSV · 2026-01-06

**Summary:**

This paper presents an improved algorithms for PatchRefiner. In overall, the authors propose to replace one of the heavy backbones with a lightweight for speed-up and also introduced a coarse-to-fine module. While some reviewers expressed some support to this paper, there is also concern on engineering contribution compared with previous version. It is noted that Reviewer k7TN marked low confidence of 2 for the comments. But when I read the paper myself, I do agree with opinion that the contribution  novelty leans engineering rather than conceptual. In fact, the main contribution is to replace one backbone with lightweight network and include a coarse-to-fine module. Both are commonly seen. But the idea to use one heavy backbone and one lightweight network can useful for the domain. Therefore, although I do agree the contribution is more toward engineer, the work sill bring some value to the domain.

**Reviewer Concerns:**

The paper received some supports from the reviewers. But the main concern on engineering contribution rather than conceptual remains which has been pointed out by Reviewer k7TN.

**Reviewer Scores:**

The reviewers have actually raised the scores.

---

### Decision · Program_Chairs · 2026-01-26

Accept (Poster)